# Megafauna mobility: Assessing the foraging range of an extinct macropodid from central eastern Queensland, Australia

**Christopher Laurikainen Gaete**[1]*, **Anthony Dosseto**[1], **Lee Arnold**[2], **Martina Demuro**[2], **Richard Lewis**[2], **Scott Hocknull**[3,4]

**1** Wollongong Isotope Geochronology Laboratory, School of Earth, Atmospheric and Life Sciences, University of Wollongong, Wollongong, New South Wales, Australia, **2** School of Physics, Chemistry and Earth Sciences, Environment Institute, and Institute for Photonics and Advanced Sensing (IPAS), University of Adelaide, North Terrace Campus, Adelaide, South Australia, Australia, **3** Geosciences, Queensland Museum, South Brisbane, Queensland, Australia, **4** School of Biological Sciences, Monash University, Melbourne, Victoria, Australia

* cgaete@uow.edu.au

## Abstract

Understanding the factors that influence the geographic range of extinct megafaunal species is crucial for reconstructing their ecology and extinction dynamics. For extant herbivores, it has been demonstrated that large body mass provides the potential for greater geographic range. Allometric scaling relationships are observed in placental mammals but have not been well-established for marsupials, in particular, extinct marsupial megafauna. Here, we employ a phylogenetic generalised least squares regression model using extant macropodids to estimate home ranges for individuals from the extinct genus *Protemnodon.* The regression model predicts a mean home range of 11.6 ± 5.8 km² This prediction, centred on Mt Etna caves, incorporates several distinct geological features with variable, known $^{87}Sr/^{86}Sr$ isotope ratios. Fossil *Protemnodon* individuals recovered from cave deposits at Mt Etna returned $^{87}Sr/^{86}Sr$ values similar to that of the host limestone, in which the cave systems formed, and the broader Mount Alma Formation. This similarity suggests that individuals foraged close to where they were fossilised, indicating a smaller home range than predicted. Smaller home ranges for individuals with a large body-mass were unexpected, attributed to a unique combination of individual behaviour, diet and/or locomotion regime within stable rainforest environments. Our results suggest that, foraging ranges in marsupial megaherbivores may be more strongly associated with environmental quality rather than body mass. New in-situ uranium-thorium and single-grain TT-OSL ages refine, and are in agreement with, previous interpretations of chronology, indicating that rainforest-adapted fauna persisted at Mt Etna until at least 280 ka. We propose that small home ranges in a stable environment, such as rainforests, predisposed these megafauna macropodids to extinction after 280ka, driven by an increasingly dry and unstable climate. Our results underscore the need for regionally specific biologies of individuals, populations and species when considering extinction pathways for Pleistocene fauna.

**Data availability statement:** All strontium, elemental concentrations, U-Th, and TT-OSL data are available in the electronic Supplementary Material. Tooth samples used for this project were intentionally selected from bulk locality collections (QML) and not individually registered because they have or will be completely lost through destructive analysis. The Queensland Museum Locality data provides opportunity for other researchers to recollect or request samples from bulk collections to undertake similar analyses. Destructive analysis of registered specimens are rarely approved.

**Funding:** SAH received funding to Queensland Museum through Project DIG for chronometric dating undertaken for this work. Funders did not play any role in study design, analyses or manuscript preparation. Funder URL: https://www.museum.qld.gov.au/collections-and-research/projects/project-dig Financial support for the TT-OSL dating research was partly provided by Australian research Council (ARC) Future Fellowship Project FT130100195 awarded to L. Arnold and FT200100816 awarded to M. Demuro. Funders did not play any role in study design, analyses or manuscript preparation. Funder URL: https://www.arc.gov.au/

**Competing interests:** The authors have declared that no competing interests exist.

## Introduction

Throughout the Quaternary, Sahul (Australia and New Guinea) was dominated by large-bodied fauna, including giant reptiles, birds, and marsupials [1,2]. Although a significant amount of research has focused on the mechanism of extinction [3–6] for Australian megafauna, less emphasis has been placed on understanding life history [2,7,8].

A key factor of life history is foraging range, i.e., how far an individual must travel to obtain necessary dietary requirements [9]. As observed with extant megafaunal herbivores, increases in body mass are positively correlated with the extent of an individual's foraging range [10,11]. Whilst body mass is considered a strong indicator of range extent in placental mammals [11], external factors including habitat type [12] may also play a role in dictating the extent of foraging range. Specifically, species residing in closed-forest habitats often have smaller foraging ranges than species of a comparable body mass residing in open-habitats [13,14]. Studies of extant macropodids have also suggested ranges are strongly associated with climatic regime and habitat productivity [15].

For extinct taxa, life histories can be inferred by using geochemical properties and biomarkers preserved in fossilised remains, in particular, fossilised enamel. Limited organic content and the greater density of enamel (compared to dentine or bone) mean isotopic signatures are less prone to alteration by diagenesis [16–18]. This relative stability ensures biologically accumulated geochemical signatures are preserved for prolonged periods of time providing an effective tool for insights into life history.

Strontium isotopes (referring to the $^{87}Sr/^{86}Sr$ ratio) in fossil enamel have been used to examine dietary foraging range in both extinct and extant organisms [19–21]. $^{87}Sr$ is produced by radioactive decay of $^{87}Rb$, thus rocks with variable Rb/Sr and age are characterised by different $^{87}Sr/^{86}Sr$ ratios [22]. Unique isotopic signatures are transferred to plants through the weathering of substrates, and into trophic chains through the consumption of plant material [23,24]. A large atomic mass and small relative difference in masses between isotopes means they are less susceptible to kinetic fractionation associated with biological processes [25]. Consequently, the strontium isotope composition of an individual reflects dietary intake and the isotopic composition of its environment [26].

For organisms, similarities between Sr and Ca allow for trace amounts of strontium to be substituted into calcium-based compounds like hydroxyapatite $[Ca_{10}(PO_4)_6(OH)_2]$ [22] providing a long-term record of biologically accumulated strontium [20]. In macropodids enamel mineralisation in incisors (I) and early molars (first and second molars, M1 & M2) coincides with weaning [27] while later forming molars (third and fourth molars, M3 & M4) post-dating weaning [27,28]. Therefore, bio-available Sr from early forming teeth will reflect dietary intake of an individual's mother, while later-forming molars reflect initial dietary independence. Using the relationship between $^{87}Sr/^{86}Sr$ ratios in geological substratum [21] and herbivores, can provide estimates of foraging range, general movement patterns, and place of origin [2,29–31]. While Sr isotopes preserved in enamel reflect an individual's foraging range during enamel mineralisation, when used in conjunction with the location of fossil deposits can provide broad inferences of total home range [32].

Fossils examined here were recovered from the western side of Mt Etna Caves National Park (formerly a limestone mine) [33,34]. Fossil assemblages suggest that between 500 – 280 thousand years ago (ka) Mt Etna was a biodiverse rainforest ecosystem, that suffered extinction when climatic shifts towards aridity saw significant faunal turnover. Rainforest species were replaced by xeric species sometime between 280 – 205 ka coinciding with an overall intensification of a drying climate [33,35]. All fossil specimens examined here are recovered from fauna interpreted to have occurred in a Middle Pleistocene rainforest palaeoenvironment, allowing us to examine for the first time the foraging range and home range of extinct rainforest macropodids.

In this study, we focus on fossilised teeth from Mt Etna Caves, focusing specifically on individuals belonging to, *Protemnodon,* a genus of a large-bodied extinct macropodids with a number of Plio-Pleistocene species [36]. *Protemnodon* is related to the extant clade of macropodines that includes *Macropus* [37]. Individual teeth are easily identified as belonging to *Protemnodon*, however, species level identifications from isolated teeth are not well delimited [36]. Body size estimates of *Protemnodon* species range between 50 – 170 kg [36], with the largest individuals significantly exceeding optimal hopping mass [38]. Elongated forelimbs and a lower crowned dentition suggested *Protemnodon* were likely browsers, using limbs to manipulate vegetation [36,39,40].

Given macropodids form a major component of Australia's fauna, understanding the past foraging ranges of extinct Quaternary species is critical, not only to better understand their life history and landscape use, but also to recover palaeobiological factors predisposing them to extinction. In this study, we used a phylogenetic generalised linear regression model to predict home ranges in species of *Protemnodon* from Mt Etna Caves based on body mass. Following this, predicted home ranges are compared to strontium isotopes in fossil teeth from Mt Etna caves, to investigate site-specific foraging range of *Protemnodon* in a Pleistocene rainforest environment. In addition, we conducted single-grain thermally transferred optically stimulated luminescence (TT-OSL) dating of the fossil-bearing sediments and in-situ uranium-thorium (U-Th) dating of fossil teeth, to refine the site chronology previously established by Hocknull et al. [33].

## Study area

Fossil macropodid specimens were sourced and identified as part of stratigraphic units collected from a series of cave deposits at Mt Etna Caves, central eastern Queensland (23°09′24″S 150°26′58″E) [35]. Large-scale stockpiling of key vertebrate deposits was undertaken while an active limestone mine operated, providing unique access to very large sedimentary deposits in two major cave systems within the limestone massif that forms the western flank of Mt Etna. Faunal remains are diverse and include a wide range of vertebrates [41], including frogs [42], reptiles [35,43], birds [35], and marsupials including large macropodids [35,44] through to small dasyurids [45,46] and murids [47,48].

Mt Etna Caves are formed within the limestone member of the Mount Etna Beds (Mount Alma Formation limestone), an allochthonous block of lower Devonian limestone and siltstones situated at the base of the Mount Alma Formation [49]. The broader Mount Alma Formation is a late Devonian conglomerate with interspersed units of fossiliferous limestone interbedded with marble (Fig 1,[50]). To the northwest, the Mount Alma Formation is bound by a Neoproterozoic to Palaeozoic intrusive unit of ultramafic rocks, tectonically emplaced during the Permian to Triassic [50]. The west of the site is characterised by a relatively complex zone with repetitive folding and faulting of Mount Alma Formation and Rockhampton Group (early Carboniferous sedimentary rock with mudstone, siltstone, felsic volcaniclastic sandstone and polymictic conglomerate) [50]. To the east, the Mount Alma Formation is bound by the Lakes Creek Formation, an early Permian arenite-mudrock, and the Alton Downs Basalt, a late Cretaceous olivine and feldspar-phyric basalt [50]. Lithology to the south is complex with a series of smaller geological units including the Lakes Creek formation, the Ellrott Rhyolite, an early Permian unit composed predominantly of volcanic and metamorphic rock, and the Sleipner Member, an early Permian unit of rudite [50]. Further afield, both the east and west of Mt Etna are bound by Quaternary alluvium. To the west, alluvium is deposited in a series of floodplain terraces associated with the Fitzroy River, derived predominantly from Devonian to Carboniferous alkaline volcanics, Permian volcaniclastic and siliciclastic rocks and basalts [51]. To the east of Mt Etna, alluvium consists of clay, stilt, sand,

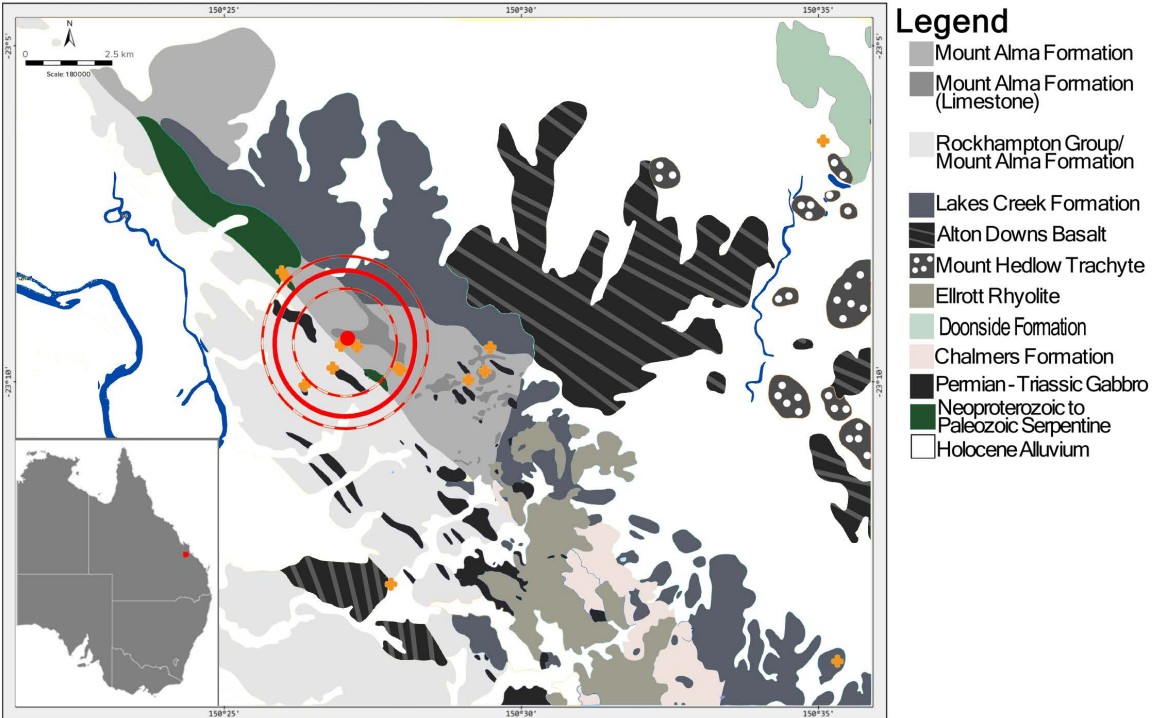

**Fig 1. Simplified geological map for the region surrounding the Mt Etna caves fossil deposits (denoted by red dot).** Locations of vegetation samples are denoted by orange cross. Estimated mean home range (11.6 ± 5.8 km² (2SE)) for species of *Protemnodon* in Mt Etna caves fossil. Red circle is representative the mean home range (11.6 ± 5.8 km² (2SE)) modelled for species of *Protemnodon* from Mount Etna Caves using a linear regression model (dashed circles denote the mean ± 2SE). Reprinted from a GeoResGlobe detailed 1:100K surface geology dataset © State of Queensland (Department of Resources) 2023.

and gravel originating as floodplain alluvium from Hedlow Creek. This creek originates in Mt Hedlow with deposits predominantly derived from Mount Hedlow Trachyte, a late Cretaceous intrusive unit composed of trachyte plugs and basalt flows.

## Materials and methods

### Modelling home ranges

Using data presented in Tucker et al. [11] and Goldingay [52], estimates of body mass (in kg) and mean home range (in km²) were collated for 17 extant species belonging to the Macropodidae family (Table A in S1 Table). For species with multiple reported estimates of body mass and/or home range, a single weighted average was calculated considering mean published values in proportion to their associated error [11]. This dataset provided the basis for a phylogenetic generalised least squares regression model [11] examining the relationship between body mass and home range in extant macropodids.

The phylogeny included in the model was based on the mammalian supertree from Upham et al. [53], which were cropped using the *NewickTreeModifier (NTM)* web tool [54] focusing solely on the Macropodidae family. Following Tucker et al. [11] a phylogenetic generalised least squares regression comparing the effect of body mass (kg) on home range (km²) was run using the *ape* 5.8 [55] and *caper* 1.0.3 [56] packages in R.

Using the regression equation from this model, home ranges (in km²) were predicted for some species of *Protemnodon*, based on current body mass estimates presented in Jones et al.

[57] and Kerr et al. [36]. Broadly, estimates of body mass range from ∼ 50 kg to a maximum 170 kg [36]. For two species defined in Kerr et al. [36], body masses range from 131 kg in *Protemnodon anak*, down to 97 kg in *Protemnodon viator*. At least three species of *Protemnodon* are recognised from Mt Etna Caves [43], although in light of the new species-level taxonomy for *Protemnodon* [41], the specific allocations require review. Despite the lack of species-level allocations, isolated teeth were grouped based on a comparison of their size to those measurements available in Kerr et al. [36]. Through this, individual teeth could be classified into two distinct size morphs: larger individuals, comparable in body mass to *Protemnodon viator* [36] (n = 3), and smaller individuals comparable in body mass to *P. otibandus/ P. tumbuna* [36] (n = 5) (Table B in S1 Table). Therefore, based on the dental remains available for this study, the body mass for Mt Etna *Protemnodon* individuals will not fall outside the range of values described above.

*Protemnodon* are not considered to be tightly analogous to extant macropodids [36,57,58] therefore home ranges in these taxa may not fall within those predicted by the linear regression using modern macropodids. However, our model provides the predicted extent of home range in a large macropodid with a body mass between 50 – 170 kg, and therefore, provides an initial test to determine how similar or different *Protemnodon* individuals were compared to extant macropodids. Our study also presents the first home range predictions for *Protemnodon* individuals that occupied a rainforest environment. Comparing predicted home range extent to foraging range inferred from Sr isotopes allows us to determine whether landscape utilisation by individuals of *Protemnodon* from Mt Etna Caves is similar or different to observed behaviour in extant macropodids. The expectation is that those observed differences will provide better explanations for their extinction during the Pleistocene, yet survival or other related macropodids.

## Sample preparation

Eight fossilised teeth (1 incisor, 7 molars, Table B in S1 Table) were identified for destructive, in-situ laser ablation analyses, representing a minimum of five individuals of *Protemnodon* from three stratigraphic units (QML1311C/D, QML1311H & QML1384LU) within two adjacent cave systems, Speaking Tube Cave and Elephant Hole Cave (S1 and S2 Figs). All necessary permits were obtained for the described study, which complied with all relevant regulations. Work undertaken at Mt Etna was by permission and collaboration with Cement Australia and Queensland Parks and Wildlife Service (Scientific Permit WITK17469216)

Given the nature of sampling, isolated teeth examined here could not be easily identified to species level, so are grouped as local *Protemnodon* from Mt Etna. While Hocknull [40] identified three taxa within these sites and associated deposits at Mt Etna, these taxa do not conform to those recently described by Kerr et al. [36] so will require further comparative work before species-level identifications are possible. Therefore, we treat the individuals as members of *Protemnodon* with clear stratigraphic and geographic similarity meaning our results represent a site-specific analysis of *Protemnodon* individuals from Mt Etna, and not a study of the *Protemnodon* genus as a whole.

Each tooth was sliced longitudinally from the crown to the base and mounted in resin for in-situ analysis by laser ablation inductively coupled plasma mass spectrometry (LA-ICP-MS).

## Luminescence dating

Two sediment sample cores (MTE17-1 and MTE17-4) were collected for luminescence dating from QML1311C/D and QML1311H. Sample core MTE17-1 was collected from massive, indurated blocks of bone-bearing sediment recovered originally from QML1311C/D during

stockpiling operations by SAH and Cement Australia in 2003-2004 (Hocknull, 2009). In-situ sediments remain on the main mine benches for QML1311H and were accessible directly for the collection of core sample MTE17-4.

Given the expected Middle Pleistocene ages of the fossil deposits [33,35], we have focussed on using single-grain quartz TT-OSL to determine when the QML1311C/D and QML1311H sediments accumulated in the Speaking Tube cave system [59,60]. This 'extended-range' luminescence dating signal exhibits considerably higher dose saturation properties than the conventional quartz OSL dating signal [61] and it has been shown to provide reliable finite depositional chronologies over Middle Pleistocene timescales at a range of independently dated sites [61–64]. Coarse-grained quartz grain fractions were extracted from the un-illuminated centres of the luminescence dating sample cores under safe light (dim red LED) conditions at the University of Adelaide's Prescott Environmental Luminescence Laboratory, and prepared for burial dose estimation using the procedures and instrumentation outlined in Arnold et al. [65]. Purified quartz grains with a diameter of 180–250 μm were carefully loaded onto aluminium discs drilled with an array of $300 \times 300$ μm holes for equivalent dose ($D_e$) evaluation. Five hundred single-grain $D_e$ measurements were made for each sample using the modified single-aliquot regenerative-dose (SAR) protocol (Table C in S1 Table), which makes use of a TT-OSL test dose to correct for sensitivity change, four preheats of 260 °C for 10 seconds in each SAR cycle, and two high temperature OSL treatments to prevent TT-OSL signal carry over from previous sensitivity-corrected regenerative dose ($L_x$) and test dose ($T_x$) measurement steps [60].The suitability of this SAR procedure is supported by reliable dose recovery test results for sample MTE17-4 (S3 Fig).

Environmental dose rates were estimated using a combination of in situ field gamma spectrometry and low-level beta counting for sample MTE17-4, and high-resolution gamma spectrometry for sample MTE17-1, taking into account cosmic ray contributions [66], an assumed minor internal alpha dose rate [67], beta dose attenuation and long-term water content. The beta and gamma dose rates of sample MTE17-1 were both determined in the laboratory using high-resolution gamma spectrometry since in situ gamma spectrometry measurements could not be undertaken at the time of sample collection. For sample MTE17-4, gamma dose rates were calculated from in situ measurements made at the original sample core position with an NaI:Tl detector (using the 'energy windows' approach; Arnold et al. [68]), while beta dose rates were calculated on dried and powdered sediment collected from within a 1 cm radius of the sample position using a Risø GM-25-5 low-level beta counter [69].

Further details of the single-grain quartz TT-OSL $D_e$ and dose rate determination procedures are provided in S1 Text, Tables C–E in S1 Table, and S3–S4 Figs.

## U-Th dating

Uranium-series dating was performed by laser ablation multi-collector inductively coupled plasma mass spectrometry (LA-MC-ICP-MS) at the Wollongong Isotope Geochronology Laboratory (WIGL), University of Wollongong. Laser ablation was performed with an ESL™ 193 nm ArF excimer laser, equipped with a TV2 cell. Samples were ablated with a laser pulse rate of 20 Hz and a fluence of 5.7 J/cm². Helium was used as a carrier gas at a flow rate of 900 mL/min, and nitrogen was added at a flow rate of 10 mL/min. Analysis was carried out with a 150 μm spot size, an ablation time of 120 s and wash out time of 30 s. Laser warm-up time was 10 s.

Thorium ($^{230}$Th, $^{232}$Th) and uranium ($^{234}$U, $^{235}$U, $^{238}$U) isotopes were measured on a Thermo Scientific™ Neptune Plus™ MC ICP-MS. All five isotopes were collected in static mode, with $^{230}$Th and $^{234}$U collected in ion counters. Helium flow rate and plasma parameters were tuned with NIST SRM 610 element standard at the start of each session, to derive

a $^{232}$Th/$^{238}$U ratio for this standard greater than 0.8 and minimise differences in fractionation between Th and U [70].

Measured $^{234}$U/$^{238}$U, $^{230}$Th/$^{238}$U and $^{232}$Th/$^{238}$U isotopic ratios were corrected for elemental fractionation and Faraday cup/SEM yield by comparing measured ratios to those of a 206 ka coral characterised independently by solution analysis. Analysis of a phosphate reference material for which isotope ratios were determined independently shows that using a coral as primary standard yields results within error of solution analyses (unpub. data). Uranium and Th concentrations were determined using NIST SRM 612 glass as calibration standard. Background subtraction and calculations of corrected ratios and concentration were performed using Iolite 4.0™ [71]. Accuracy was assessed using a 124 ka coral (MK16) also characterised independently by solution analysis. Results yield ($^{234}$U/$^{238}$U) and ($^{230}$Th/$^{238}$U) ratios within error of values determined by solution analysis (1.110 ± 0.002 and 0.764 ± 0.007, respectively).

Analyses were conducted along transects perpendicular to the teeth's surface. For each transect, we calculated an open-system U-Th age, using R package *UThwigl* [72] based on the iDAD model of Grün et al. [73], Sambridge et al. [74].

## Strontium isotope analysis

Strontium isotope ratios were measured by LA-MC-ICP-MS at WIGL, using the same laser ablation system and MC-ICP-MS as for U-series dating (see above). Samples were ablated with a laser pulse rate of 10 Hz and a fluence of 8.5 J/cm$^2$. Helium was used as a carrier gas at a flow rate of 800 mL/min, and nitrogen added at a flow rate of 10 mL/min. Analysis was carried out with a 150 μm spot size, an ablation time of 50 sec and wash out time of 30 sec. Given enamel mineralisation in brachydont teeth occurs sequentially from the crown to the root of the tooth, strontium isotopes were analysed along this axis to examine intra-tooth variations in $^{87}$Sr/$^{86}$Sr [75]. For each tooth, six transects of ablation spots were undertaken along these axes for both enamel and dentine. The MC-ICP-MS was equipped with a jet sample cone and a x skimmer cone. Strontium ($^{84}$Sr,$^{86}$Sr, $^{87}$Sr, $^{88}$Sr), rubidium ($^{85}$Rb) and krypton ($^{82}$Kr, $^{83}$Kr) isotopes were collected in Faraday cups in static mode.

At the start of each session, the system was tuned with SRM NIST 610 glass with a spot size of 65 μm, scan speed of 5 μm/sec., laser pulse rate of 5 Hz and fluence of 6.7 J/cm$^2$. Before and after each sample measurement, 3 spots were ablated on a fragment of clam shell and on a section of seal tooth enamel, both assumed to have the modern seawater $^{87}$Sr/$^{86}$Sr ratio (0.709182, Miller and Kent [76]). The clam shell was used a calibration standard and the seal tooth enamel to assess accuracy of measurements.

Data reduction was performed in iolite 4.0™ [71], using the 'Sr Isotopes' data reduction scheme. Baseline was subtracted from all isotopes. $^{87}$Sr/$^{86}$Sr ratios were first corrected internally for each standard and sample. $^{88}$Sr, $^{86}$Sr and $^{84}$Sr were corrected from isobaric interferences using $^{82}$Kr, and $^{87}$Sr using $^{85}$Rb. The measured $^{88}$Sr/$^{86}$Sr ratio was used to calculate a mass bias factor, then applied to the $^{87}$Sr/$^{86}$Sr, $^{84}$Sr/$^{86}$Sr and $^{87}$Rb/$^{86}$Sr ratios corrected for isobaric interferences. The internally corrected $^{87}$Sr/$^{86}$Sr ratio of samples and seal tooth enamel were then further corrected by using the internally corrected $^{87}$Sr/$^{86}$Sr ratio of the clam shell and a standard-sample-standard bracketing method [20,77]. Precision and accuracy were assessed using the seal tooth enamel. Except for one session, $^{87}$Sr/$^{86}$Sr isotopic ratios were within error of the modern seawater value (laboratory average 0.709212 ± 0.000006 2SE).

## Dietary foraging range estimates

Estimates of foraging range were based on predicted $^{87}$Sr/$^{86}$Sr variability in local geology [2,29], focused on movement from limestone to adjacent geological substrates. Underlying

geological features were obtained from a detailed 1:100K surface geology dataset (Queensland Government, GeoResGlobe). To examine variations in Sr isotopes across geological substrates, $^{87}Sr/^{86}Sr$ isotope ratios were measured in 24 plant samples representing 11 distinct geological features encapsulating a 20 km x 25 km region around Mt Etna (Fig 1, S1 File, Table F in S1 Table). As bio-available $^{87}Sr/^{86}Sr$ is associated with underlying geology [23,24], using this plant dataset, movements in *Protemnodon* could be defined as localised (constrained to the Mount Alma Formation limestone and/or Mount Alma Formation) or reflect larger scale movement to adjacent geological substrates.

## Results

### Modelling home range

Body mass accounted for 34% of the variation observed in macropodid home range ($R^2$ = 0.34, F (1, 15) = 9.166, p = 0.008) (Fig 2). The linear regression equation in the Macropodidae model (logY = 1.4logX – 1.82) is steeper than the terrestrial herbivore model [11] suggesting proportionally greater increases in home range driven by body mass. For the *Protemnodon* genus, mean home range incorporating all estimates of body mass is 11.6 ± 5.8 km². Estimates range between 3.6 km² for a 50 kg individual to 19.8 km² for the maximum hypothesised body mass (170 kg). When considering individual species defined in Kerr et al. [36], predicted home ranges are variable, with 13.8 km² for *P. anak* (131 kg), and 9.1 km² for *P. viator* (97 kg).

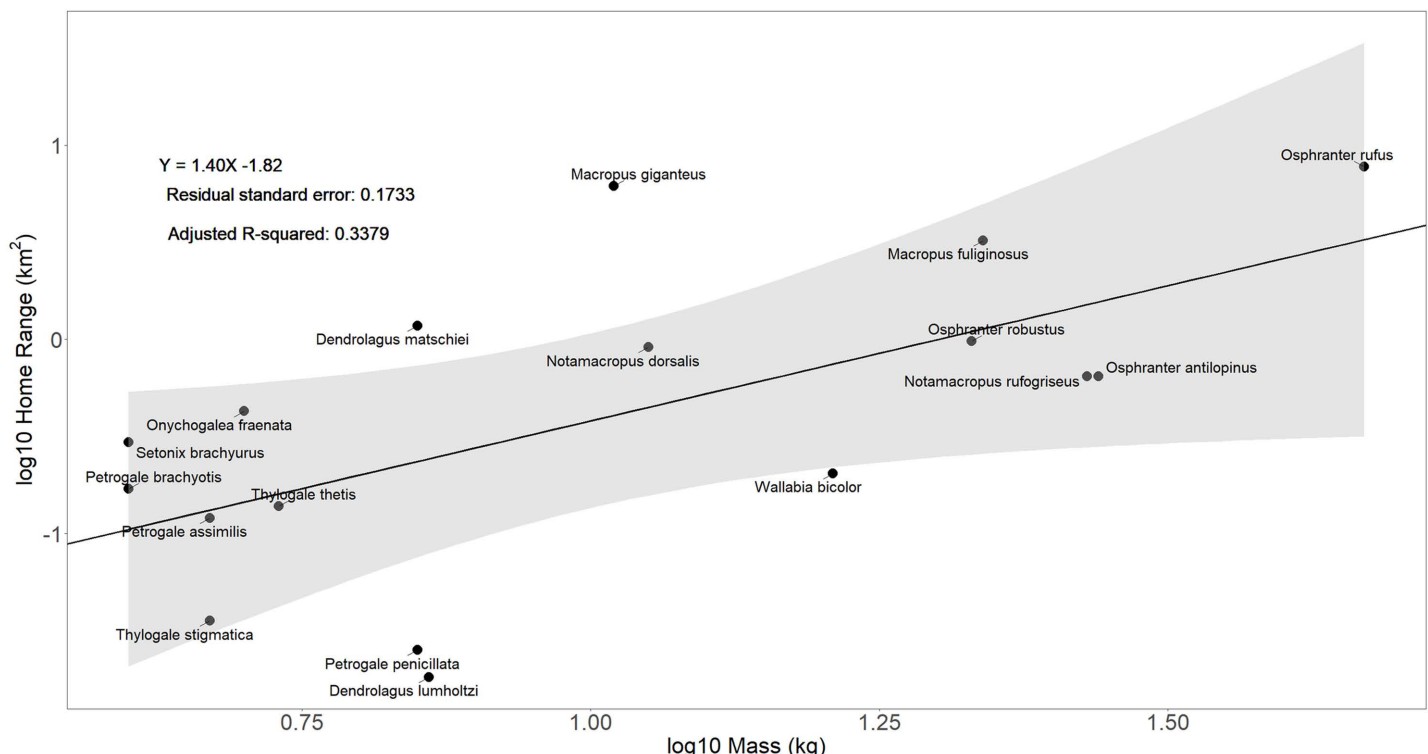

**Fig 2. Home range as a function of species body mass for extant members of the family Macropodidae.** Solid black line represents the phylogenetic linear regression undertaken for all extant Macropodidae.

## Luminescence dating

The single-grain TT-OSL $D_e$ distributions of samples MTE17-1 and MTE17-4 display limited scatter and contain single dose populations, with the majority of individual $D_e$ values falling within the 95% C.I. of the weighted mean $D_e$ values (Fig 3; grey shaded bands). Both samples exhibit low to moderate overdispersion values (3–30%), which are similar to those reported for well-bleached and unmixed single-grain TT-OSL $D_e$ datasets elsewhere [65,78–82]. Neither $D_e$ dataset is considered significantly positively skewed according to the weighted skewness test outlined by Bailey and Arnold [83] and [84]. Application of the maximum log likelihood ($L_{max}$) test [85] indicates that the central age model (CAM) is statistically favoured over the three- or four-parameter minimum age models (MAM-3 or MAM-4) of Galbraith et al. [86] for both datasets. These $D_e$ characteristics suggest that the samples were not significantly affected by partial bleaching, syn-depositional reworking of sediments within the cave system or post-depositional complications (e.g., sediment mixing or beta dose rate heterogeneity). Consequently, representative single-grain TT-OSL burial dose estimates have been calculated using the weighted mean (CAM) $D_e$ estimate.

Sample MTE17-1 from unit QML 1311 C/D yields a single-grain TT-OSL age of 291.5 ± 28.3 ka (1σ), while sample MTE17-4 from unit QML 1311H provides an age of 304.2 ± 26.1 ka (1σ) (Table 1). These two TT-OSL ages are statistically indistinguishable at 1σ and suggest that the QML 1311 C/D and QML 1311H sediments were most likely deposited during MIS 8 to 9, with the full 2σ TT-OSL uncertainty ranges encompassing the period spanning late MIS 10 to early MIS 7.

## Open-system U-Th dating

Open-system U-Th ages range from 210 + 13/-13 ka (WIGL8544) to 280 + 21/-18 ka (WIGL8550). For each tooth, both transects return ages within error of each other, except for incisor WIGL8549 (Table 2). For stratigraphic unit QML1311H, ages vary between 210

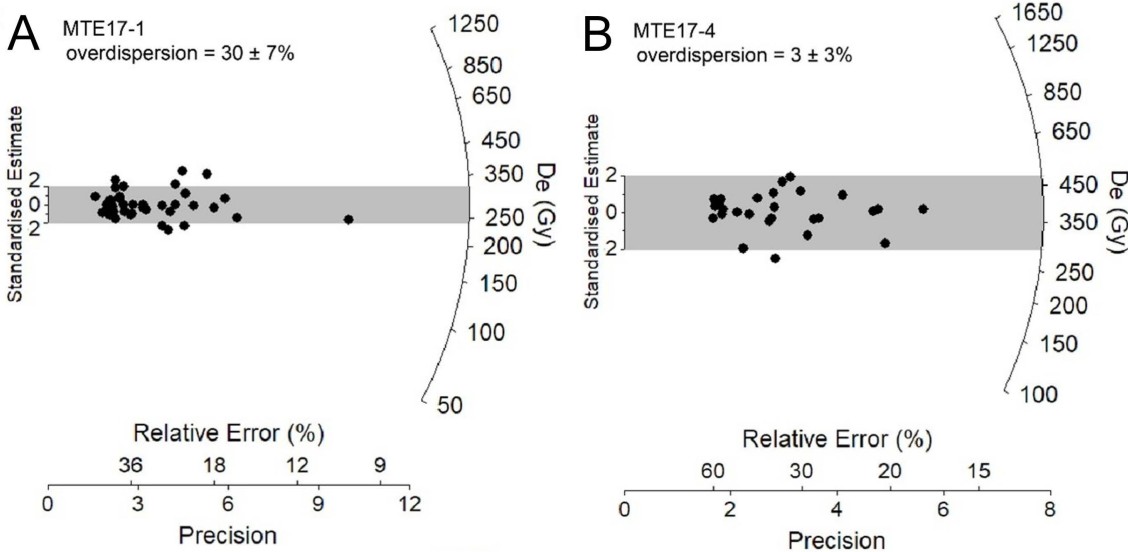

**Fig 3. Single-grain $D_e$ distributions for the TT-OSL dating samples, shown as radial plots.** The grey bands on the radial plots are centred on the $D_e$ values used for the age calculations, which were derived using the central age model (CAM). Individual $D_e$ values that fall within the shaded region are consistent with the CAM burial doses used for age calculation at 2σ.

**Table 1. Summary of single-grain TT-OSL dating results for the QML1311C/D and QML1311H sediment samples.**

| Sample | Stratigraphic Unit | Grain size (μm) | Water content (%)[a] | Environmental dose rate (Gy/ka) | | | | | Equivalent dose ($D_e$) data | | | Age (ka)[g,k] |
|---|---|---|---|---|---|---|---|---|---|---|---|---|
| | | | | Beta dose rate[b,c] | Gamma dose rate[b,d] | Cosmic dose rate[e] | Internal dose rate[f] | Total dose rate [g] | No. of grains or aliquots[h] | Over-dispersion (%)[i] | $D_e$ (Gy)[g,j] | |
| MTE17-1 | QML1311C/D | 180–250 | 5/ 20 | 0.50 ± 0.03 | 0.42 ± 0.02 | 0.01 ± 0.01 | 0.03 ± 0.01 | 0.95 ± 0.06 | 40/ 500 | 30 ± 7 | 277.4 ± 19.3 | 291.5 ± 28.3 |
| MTE17-4 | QML1311H | 180–250 | 16/ 16 | 0.71 ± 0.04 | 0.49 ± 0.02 | 0.01 ± 0.01 | 0.03 ± 0.01 | 1.23 ± 0.07 | 27/ 500 | 3 ± 3 | 375.5 ± 23.2 | 304.2 ± 26.1 |

[a]Present-day water content/ long-term estimated water content, expressed as % of dry mass of mineral fraction, with an assigned 1σ relative uncertainty of ± 20%. The present-day water content of MTE17-4 is considered representative of long-term conditions. The long-term water content of MTE17-1 is taken as 35% of the saturated water content.

[b]Beta and gamma dose rates of sample MTE17-1 were determined using high-resolution gamma spectrometry. *In situ* gamma spectrometry measurements could not be undertaken on this sample during sample collection. Specific activities and radionuclide concentrations have been converted to dose rates using the conversion factors given in Guérin et al. [87], making allowance for beta dose attenuation [88,89].

[c]Beta dose rate of sample MTE17-4 was calculated on dried and powdered sediment using a Risø GM-25-5 low-level beta counter [69]. Radionuclide concentrations and specific activities of beta counting standards have been converted to dose rates using the conversion factors given in Guérin et al. [87].

[d]Gamma dose rate of sample MTE17-4 was calculated from *in situ* measurements made with an NaI:Tl detector, using the 'energy windows' approach [68,90]. Radionuclide concentrations and specific activities of gamma spectrometry calibration materials, and K, U, Th concentrations determined from the field gamma-ray spectra have been converted to dose rates using the conversion factors given in Guérin et al. [87].

[e]Cosmic-ray dose rates were calculated according to Prescott and Hutton [66] and assigned a relative 1σ uncertainty of ± 10%.

[f]The assumed internal alpha + beta dose rate for quartz, with an assigned relative 1σ uncertainty of ± 30%, is based on intrinsic $^{238}U$ and $^{232}Th$ contents published by Mejdahl [91], Bowler et al. [67], Jacobs et al. [92], Pawley et al. [93], and Lewis et al. [94], and an a-value of 0.04 ± 0.01 [95,96]. Intrinsic radionuclide concentrations and specific activities have been converted to dose rates using the conversion factors given in Guérin et al. [87], making allowance for beta dose attenuation due to grain-size effects [88].

[g]Mean ± total uncertainty (68% confidence interval), calculated as the quadratic sum of the random and systematic uncertainties.

[h]Number of $D_e$ measurements that passed the SAR rejection criteria and were used for $D_e$ determination/ total number of $D_e$ values analysed.

[i]The relative spread in the $D_e$ dataset beyond that associated with the measurement uncertainties for individual $D_e$ values.

[j]Sample-averaged $D_e$ value for each sample, calculated using the central age model (CAM) of Galbraith et al. [86].

[k]Total uncertainty includes a systematic component of ± 2% associated with laboratory beta-source calibration.

**Table 2. Open-system U-Th ages for individual transect analyses completed on fossil *Protemnodon* teeth.**

| Tooth ID | Stratigraphic Unit | Tooth type | Open-system U-Th age (ka) | +2SD (ka) | -2SD (ka) |
|---|---|---|---|---|---|
| WIGL8543_1 | QML1311H | Molar | 234 | 9 | 6 |
| WIGL8543_2 | QML1311H | Molar | 255 | 50 | 15 |
| WIGL8544_1 | QML1311H | Molar | 212 | 19 | 21 |
| WIGL8544_2 | QML1311H | Molar | 208 | 17 | 16 |
| WIGL8545_1 | QML1311H | Molar | 230 | 20 | 17 |
| WIGL8545_2 | QML1311H | Molar | 245 | 23 | 18 |
| WIGL8546_1 | QML1311H | Molar | 480 | 190 | 140 |
| WIGL8546_2 | QML1311H | Molar | 260 | 5 | 20 |
| WIGL8547_1 | QML1384LU | Molar | 264 | 12 | 14 |
| WIGL8547_2 | QML1384LU | Molar | 291 | 29 | 30 |
| WIGL8548_1 | QML1384LU | Molar | 228 | 29 | 25 |
| WIGL8548_2 | QML1384LU | Molar | 213 | 13 | 9 |
| WIGL8549_1 | QML1311C/D | Incisor | 240 | 5 | 6 |
| WIGL8549_2 | QML1311C/D | Incisor | 211 | 6 | 4 |
| WIGL8550_1 | QML1311C/D | Molar | 271 | 35 | 30 |
| WIGL8550_2 | QML1311C/D | Molar | 285 | 27 | 23 |

For each specimen, two transects have been ablated.

For WIGL8546_1 date is significantly higher with

[a]large error – thus this analysis was excluded

+13/-13 ka and 267 +48/-20 ka; for unit QML1384LU, they vary between 215 +12/-8 ka and 268 +11/-12 ka; and for unit QML1311C/D, between 222 +4/-3 ka and 280 +21/-18 ka. There is no significant difference between ages across stratigraphic units (one way analysis of variance (Anova), p = 0.829). Oldest minimum ages for QML1311H, QML1384LU and QML-1311C/D are 260 +5/-20 ka, 291 +29/-25 ka, and 285 +27/-23 ka respectively (Table 2).

## Strontium isotopes

$^{87}$Sr/$^{87}$Sr ratios range from 0.70667 ± 0.00008 (2SE internal analytical uncertainty) to 0.70932 ± 0.00046 (2SE). For most teeth, intra-sample variability is minimal with little to no variation across individual ablated transects (S5 Fig). However, WIGL8548 exhibits notable intra-tooth variation, with $^{87}$Sr/$^{86}$Sr ratios ranging between 0.707474 ± 0.000072 (2SE) and 0.709319 ± 0.000046 (2SE) (Fig 4). Mean $^{87}$Sr/$^{87}$Sr ratios present in the crown of the molar, 0.708804 ± 0.000216 (2SE), are significantly higher than those measured in the base of the tooth (0.707706 ± 0.000086 (2SE)) (Welch Two Sample t-test, p < 0.001).

For stratigraphic unit QML1311H, $^{87}$Sr/$^{86}$Sr ratios significantly differ between individual teeth (Kruskal-Wallis test, p < 0.001). Pairwise comparisons using Dunn's test indicate $^{87}$Sr/$^{86}$Sr ratios in WIGL8544 are significantly higher than those observed in WIGL8543, WIGL8545, and WIGL8546 (p < 0.001) (Fig 5). For remaining specimens in this stratigraphic unit (WIGL8543, WIGL8545, and WIGL8546), $^{87}$Sr/$^{86}$Sr values are similar. In stratigraphic unit QML1384LU, $^{87}$Sr/$^{86}$Sr ratios do not vary between the two individual *Protemnodon* specimens: WIGL8547 (0.707841 ± 0.000136), and WIGL8548 (0.708105 ± 0.000058) (Welch Two Sample t-test, p = 0.063). In QML1311C/D, WIGL8549 displays an $^{87}$Sr/$^{86}$Sr of 0.707820 ± 0.000113 (2SE), whereas WIGL8550 exhibits a significantly lower $^{87}$Sr/$^{86}$Sr ratio of 0.706855 ± 0.000020 (2SE) (Welch Two Sample t-test, p < 0.001). This low $^{87}$Sr/$^{86}$Sr is present in all transects analysed across WIGL8550.

$^{87}$Sr/$^{87}$Sr ratios are variable between stratigraphic units: ranging from 0.70667 ± 0.00008 (2SE) to 0.70837 ± 0.00010 (2SE) in QML1311C/D, 0.70923 ± 0.00020 (2SE) to 0.70923 ± 0.00018 (2SE) in QML1311H, and 0.70747 ± 0.00007 (2SE) to 0.70931 ± 0.00005 (2SE) in QML1384LU. Comparing *Protemnodon* enamel to local geology, mean $^{87}$Sr/$^{86}$Sr isotope ratios in all four individuals from QML1311H show values overlapping with the mean $^{87}$Sr/$^{86}$Sr ratios measured in Mount Alma Formation limestone (0.708514 ± 0.000348, 2SE, n = 4), however,

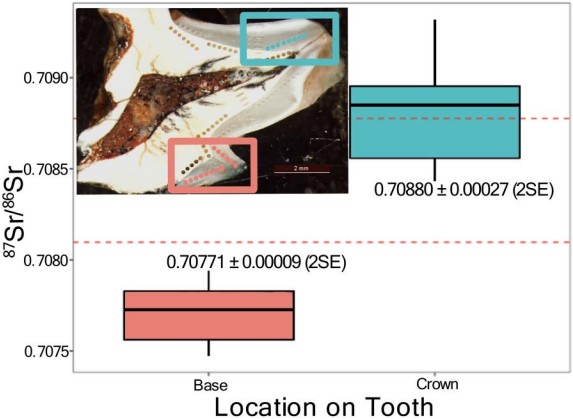

**Fig 4. $^{87}$Sr/$^{86}$Sr isotopic ratios as a function of sampling location for Protemnodon specimen WIGL8548.** Location of individual enamel ablation transects undertaken on the molar cross with two transects ablated towards the base of the tooth (red) and an additional transect towards the crown (blue).

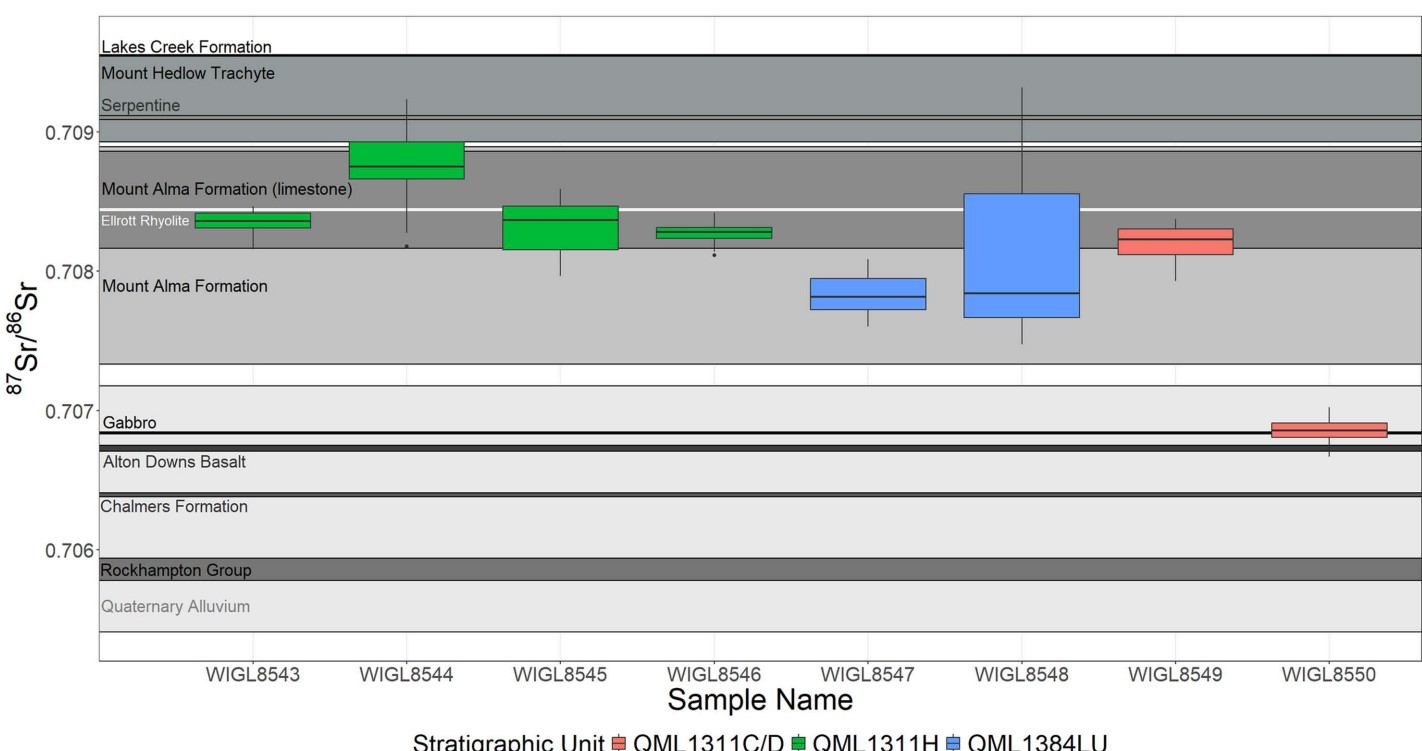

**Fig 5. $^{87}$Sr/$^{86}$Sr isotopic ratios across individual Protemnodon specimens.** Sr isotope ratios in *Protemnodon* are compared against the mean (± 2SE) $^{87}$Sr/$^{86}$Sr isotope ratios measured in vegetation samples reflecting underlying geology (Table F in S1 Table).

in WIGL8544, higher isotope ratios measured in some transects overlap with values measured in Mount Hedlow Trachyte (0.709240 ± 0.000311, 2SE, n = 2), and Neoproterozoic to Paleozoic units of Serpentine (0.709102 ± 0.000014, 2SE, n = 1), and $^{87}$Sr/$^{86}$Sr isotope ratios in WIGL8543 and WIGL 8545 show minor overlap with those reported in Ellrott Rhyolite (0.708443 ± 0.000007, 2SE, n = 1) (Fig. 5). For the two samples from QML1384LU, while measured $^{87}$Sr/$^{86}$Sr isotope ratios are outside values measured in the Mount Alma Formation limestone, they still fall within the range of values measured in the Mount Alma Formation (0.708138 ± 0.000572, 2SE, n = 6), the broader formation encapsulating the local limestone beds (Figs 1, 5). WIGL8548 also shows minor overlap with Sr isotope ratios measured in Ellrott Rhyolite (0.708443 ± 0.000007, 2SE, n = 1) (Fig 5). In QML1311C/D, while $^{87}$Sr/$^{86}$Sr isotope ratios in WIGL8549 are similar to those reported in the Mount Alma Formation and Mont Alma Formation limestone, the Sr isotope composition of WIGL8550 is significantly lower, with values similar to the Alton Downs Basalt (0.706728 ± 0.000022, 2SE, n = 1), Permian to Triassic Gabbro (0.706839-0.000008, 2SE, n = 1), and alluvial deposits associated with the Fitzroy River (0.706292 ± 0.000883, 2SE, n = 4) (Fig 5).

## Discussion

### Refined chronology of Mt Etna caves deposits

Previous site chronology defined in Hocknull et al. [33] assigned QML1311C/D a maximum age of < 330 ka based on stratigraphic position and dating of a basal flowstone (ROK04/04, 326 ± 22 ka) associated with the underlying QML1311F unit. The new TT-OSL age of 291.5 ± 28.3 ka (1σ) for QML1311C/D agrees with this maximum age, while the open-system U-Th

ages of *Protemnodon* teeth are somewhat younger, with ages ranging between 222 + 4/-3 ka and 280 + 21/-18 ka (2σ). Collectively, new ages for QML1311C/D support existing chronology established in Hocknull et al. [33] with the oldest U-series age (280 + 21/-18 ka (2σ)) now providing a minimum age suggesting fossil deposition occurred between ~ 280 – 330 ka.

For QML1311H, Hocknull et al. [33] reported a range of ages based on U/Th dating of macropodid bones and calcite infills, with bone ages ranging from 216 ± 4 ka to 285 ± 7 ka and calcite infill ages ranging from 125 ± 4 ka to 454 ± 48 ka. These vastly different ranges underscore the complexity of QML1311H deposit relative to other deposits within the system as noted by Hocknull [40]. The position of QML1311H relative to QML1311F, and QML1311 CD, indicate that the QML1311H deposit has infilled below a lithified and undercut older deposit, QML1311F [40]. Therefore, the range of ages is not surprising since we would expect to find reworked (older) bones within QML1311H. In our revised chronology, we consider direct dating of sediments using TT-OSL in combination with more precise laser ablation methods directly dating fauna. Here, our TT-OSL age for the QML1311H sediment deposit is 304.2 ± 26.1 ka (1σ) and open-system U-Th dating of *Protemnodon* teeth yield ages between 208 + 17/-16 ka and 260 + 50/-20 ka (2σ), younger than the previous estimate from calcite infills but consistent with previous U/Th ages from macropodid bone. Considering additional ages presented in supplementary material from Hocknull et al. [33] (Table G in S1 Table) are consistent with the new U-series ages, we propose that this older minimum age of 454 ± 48 ka may be an anomaly potentially reflecting a reworked bone with calcite, originating from an older deposit, likely QML1311F.

Excluding this much older and potentially reworked age, and using the second oldest minimum age (285 ± 7 ka) from Hocknull et al. [33] along with our direct sediment age using TT-OSL, we conclude that the majority of deposition of QML1311H occurred between 285 ± 7ka – 304 ± 26 ka. This age range overlaps with QML1311 C/D of around 280-330 ka. The potential impact of reworked fossils has been considered by Hocknull [40] and this supports the use of direct-dating methods for fauna when considering specific chronometric age-related questions. Ultimately, the refined age for QML1311H is younger than that previously defined in Hocknull et al. [33]. Similar ages in QML1311H and QML1311C/D are supported by lithological similarity, depositional context, and similar faunal assemblages [40] suggesting these two deposits accumulated over a similar period of time.

For QML1384LU, Hocknull et al. [33] suggested minimum ages of 267 ± 5 ka based on macropodid bone, and 332 ± 14 ka from calcite infills. In the present study, open-system U-Th dating yields ages between 213 + 13/-9 ka and 291 + 29/-30 ka (2σ), both in agreement with the previous minimum ages established in Hocknull et al. [33]. However, given TT-OSL dates were not obtained, we are unable to further constrain the chronology, meaning the oldest minimum age of 332 ± 14 ka remains, with the minimum age of QML1384LU being greater than the ages for QML1311H and QML1311C/D.

A detailed review of the geochronology of the fossil deposits at Mt Etna is ongoing, incorporate new ages and depositional observations that will further refine the chronology of fauna. Undertaking this level of chronological review was outside the scope of the present study, which is focused on home ranges of megafauna. However, the chronology presented here at least provides evidence that the individuals being assessed existed in a similar period of time. Bootstrap cluster analyses of faunal assemblages grouped the three stratigraphic units with tropical rainforest assemblages in New Guinea and Australia's tropics, whilst a single younger deposit (QML1312, 170 – 205 ka) clustered with central Australian xeric habitats [33]. Additional TT-OSL dating and U-Th dating of fossil teeth from younger deposits is necessary to better understand the speed of transition from rainforest to xeric habitat, however, our current results support conclusions drawn in Hocknull et al. [33] suggesting rainforest conditions

persisted until around 280 ka with the local extinction of *Protemnodon* occurring shortly after this time.

## Foraging range

Understanding foraging range of extinct organisms is crucial for reconstructing past ecosystems and habitat utilisation. Phylogenetic linear regressions for Macropodidae suggest body mass accounts for some variation observed in home range trends (Fig 2), however, allometric relationships were significantly weaker than those observed in terrestrial herbivores from Tucker et al. [11]. Considering extant outliers (specimens falling outside the 99% confidence interval shown in Fig 2, Table A in S1 Table), limited home ranges in *Dendrolagus lumholtzi* are attributed to their arboreal nature, and highly fragmented habitat [97,98], while smaller ranges in *Petrogale penicillata* are driven by attachment to diurnal refugia, and habitat productivity associated with their tropical environment [99]. Conversely, in *Dendrolagus matschiei*, larger home ranges are attributed to a continuous rainforest habitat, and low productivity associated with their higher altitude habitat [9,100]. In *Macropus giganteus*, larger observed foraging ranges (relative to those predicted by body mass) are attributed to larger scale, male biased dispersal [101] with Fisher and Owens [15] suggest variations in range size and social organisation are more strongly correlated with variations in habitat productivity, and less influenced by body size. Therefore, it is conceivable that, although body mass may account for some variability in macropodid home range, ranges may also be influenced by external factors including macropodid behaviour and the environment they reside in.

Based on the linear regression model, mean predicted home range for species of *Protemnodon* is $11.6 \pm 5.8$ km$^2$ with a maximum range of 19.8 km$^2$ for a 170 kg individual. For two species of *Protemnodon* described in Kerr et al. [36] predicted ranges exceed 9 km$^2$. In the context of the Mt Etna local region, a $11.6 \pm 5.8$ km$^2$ home range would span several geological substrates including the Mount Alma Formation, Rockhampton Group, Lakes Creek formation and smaller units of Serpentine and Gabbro (Fig 1).

As demonstrated by Funck et al. [21], enamel $^{87}$Sr/$^{86}$Sr ratios, used in conjunction with a bio-available strontium isoscape can be used to spatially predict foraging ranges with a relatively high degree of accuracy. Trace elements and Rare Earth Element (REE) concentrations of fossil enamel show that specimens included here have not been significantly affected by diagenesis, and therefore, the Sr isotope composition of teeth reflects bioaccumulation of strontium (S1 File, S6–S11 Figs). Despite minor variations between specimens, five samples clearly show $^{87}$Sr/$^{86}$Sr ratios within the range of values reported in the Mount Alma Formation limestone (Fig 5). In two of the remaining specimens, while $^{87}$Sr/$^{86}$Sr isotope ratios fall outside the range of values seen in the Mount Alma Formation limestone, they are still within the range of values for the broader Mount Alma Formation (Fig 5). The incorporation of Sr within the teeth of the *Protemnodon* individuals occurred early in life during tooth development, yet the worn nature of several of the teeth indicate considerable time between tooth formation and death. Therefore, the overall $^{87}$Sr/$^{86}$Sr ratio signal from the teeth matching closely the $^{87}$Sr/$^{86}$Sr ratio of the local limestone, and broader Mount Alma Formation indicates that (1) during enamel mineralisation individuals exhibited small foraging ranges and (2) individuals did not move significant distances between enamel mineralisation and death. Comparing Sr isotope values in these seven specimens to the surrounding geology, significantly higher $^{87}$Sr/$^{86}$Sr isotope ratios are observed in the Lakes Creek Formation (0.709549 ± 0.000006, 2SE, n = 1), Mount Hedlow Trachyte (0.709240 ± 0.000311, 2SE, n = 2) and smaller units of Serpentine (0.709102 ± 0.000014, 2SE, n = 1). Additionally, $^{87}$Sr/$^{86}$Sr isotope ratios in the Rockhampton Group (0.705858 ± 0.000081, 2SE, n = 2), Alton Downs Basalt (0.706728

± 0.000022, 2SE, n = 1), Chalmers Formation (0.706395 ± 0.000014, 2SE, n = 1), Quaternary Alluvium (0.706292 ± 0.000883, 2SE, n = 4) and Permian to Triassic units of Gabbro (0.706839-0.000008, 2SE, n = 1) are all noticeably lower than $^{87}Sr/^{86}Sr$ isotope ratios measured in the Mount Alma Formation (Fig 5). This suggests that the $^{87}Sr/^{86}Sr$ values observed for most of the Mt Etna *Protemnodon* individuals reflect short scale foraging movements, mostly constrained to the local Mount Alma Formation limestone, and Mount Alma Formation. In WIGL8544, higher isotope ratios measured in some transects overlap with values measured in Mount Hedlow Trachyte (0.709240 ± 0.000311, 2SE, n = 2), and Neoproterozoic to Paleozoic units of Serpentine (0.709102 ± 0.000014, 2SE, n = 1) potentially reflecting movements beyond the local Mount Alma Formation (Figs 1 and 5). However, the $^{87}Sr/^{86}Sr$ isotope ratios returned in most transects mirror Sr values found in the local Mount Alma Formation limestone and broader Mount Alma Formation. While similar isotope ratios are reported in Ellrott Rhyolite (0.708443 ± 0.000007, 2SE, n = 1), this occurs to the south of Mount Etna Caves interspersed between the Lakes Creek Formation, Permian to Triassic Gabbro and the Rockhampton Group (Fig 1). Therefore, although it could be conceivable Sr isotope composition of individual *Protemnodon* could reflect foraging over Ellrott Rhyolite, unless individuals were constrained entirely to Ellrott Rhyolite – unlikely given only small intrusive units are exposed (Fig 1) – it is likely Sr composition of teeth would also reflect significantly lower isotope ratios derived from adjacent features including Lakes Creek Formation, Permian to Triassic Gabbro and the Rockhampton Group. Moreover, if $^{87}Sr/^{86}Sr$ isotope ratios in *Protemnodon* are interpreted as a 'southern' foraging range constrained to the Ellrott Rhyolite, this would suggest all individuals were non-vagile during enamel mineralisation, but then undertook a one-way journey > 7 km to become a part of the Mt Etna fossil record. Given similar $^{87}Sr/^{86}Sr$ isotope ratios are observed in older individuals from QML1384LU (> 330 ka) and younger individuals (280 – 330 ka), we suggest continuous, long-term, unidirectional range shifts across all individuals prior to death are unlikely, therefore Sr isotopic composition is more likely to reflect a 'local' foraging range. Ultimately, a more extensive surveying of bio-available strontium and an isoscape quantifying spatial variability of $^{87}Sr/^{86}Sr$ is required to predict foraging ranges with a high degree of spatial accuracy [21]. However, significantly higher and lower Sr values in surrounding substrates indicate that most *Protemnodon* individuals found as fossils at Mt Etna Caves were likely non vagile, with movements likely constrained to the local, Mount Alma Formation in 7 specimens, and further constrained to local, limestone outcrops in 5 specimens (Fig 5).

Limited home range in most *Protemnodon* individuals from the Mt Etna Caves deviate from that predicted using our linear regression model based on body mass (11.6 ± 5.8 km², Fig 1). This may suggest that like modern macropodids, the foraging ranges of *Protemnodon* from Mt Etna are not entirely dictated by body mass, and may be more strongly influenced by external factors including behaviour and environment [15]. These findings support previous interpretations of the *Protemnodon* genus from Kerr et al. [36] suggesting ecomorphological variations across species were indicative of a broad array of ecological adaptations (including locomotory variations) across different palaeoenvironments. Therefore, the extinction risks for *Protemnodon* during the Pleistocene cannot be simplified using broad, genus-level interpretations. Instead, regionally specific baseline biologies need to be built using objective independent techniques.

Considering the current understanding of the genus, we propose that the limited foraging ranges at Mt Etna for *Protemnodon*, was driven by (1) a stable resource available surrounding the caves (rainforest), (2) dietary preferences of the individuals and/or, (3) unique locomotory biomechanics, that combined, limited the terrestrial dispersal capacity of the Mt Etna *Protemnodon*.

Whilst increasing body mass is often conversely related to foraging range, Tucker et al. [11] noted that the extent of home ranges may be influenced by the environment an individual resides in. Studies of extant organisms suggest inverse relationships exist between resource availability and foraging range [102–105]. Namely, if an ecosystem is productive, organisms can obtain all necessary dietary requirements within a limited foraging envelop and are therefore considerably less mobile than individuals in resource-limited environments. These trends have been observed in west Amazonian closed forest herbivores [13], and small to large frugivorous ungulates in the Ituri forest, Democratic Republic of Congo [106]. As home ranges of extant macropodids are strongly associated with climate [15], we suggest that climatic stability associated with Mt Etna's rainforest environment [33] provided a productive ecosystem where dietary needs could be met with relatively small foraging ranges. Therefore, the limited movement suggested by $^{87}Sr/^{86}Sr$ ratios in Mt Etna *Protemnodon* may reflect an abundance of resources adjacent to fossil-bearing caves. These conclusions further support notions where unlike eutherians mega-herbivores, home ranges of macropodids may be more strongly associated with climatic and environmental conditions rather than body mass [15].

Rainforest conditions, as demonstrated by the vertebrate fauna at Mt Etna, were present at least 500 ka through to approximately 280 ka [33]. This represents approximately 200,000 years of persistent closed forest conditions at Mt Etna. Therefore, we conclude that the local resources needed for individual *Protemnodon* at Mt Etna were likely met over this long period of time prior to major habitat changes associated with aridity [33].

A second potential explanation for limited foraging ranges is differing dietary preferences between *Protemnodon* and extant *Macropus*. While *Macropus* are predominantly grazers [107], the presence of elongated forelimbs [39], low-crowned molars [37,39], and stable isotope analyses [108,109] suggest *Protemnodon* species were largely browsers. Lower crowned molars observed in Mt Etna *Protemnodon* support these browsing dietary preferences [40], however, more detailed analysis is needed. As browsers can attain dietary requirements within smaller foraging ranges than grazers of a comparable size [14], it is conceivable that restrictive foraging range in Mt Etna *Protemnodon* – compared to extant *Macropus* – are attributed to dietary preference. While *Macropus* require large areas for grazing, it is likely that the Mt Etna *Protemnodon* could obtain browse resources within a significantly smaller area, especially within a dominantly forested limestone environment where grasses seldom grow.

A third alternative or complimentary explanation for limited foraging ranges observed in species of *Protemnodon* may be due to differences in movement biomechanics. All extant *Macropus* are bipedal hoppers, well equipped for efficient long-distance traversal [39,110]. However, with body mass estimates exceeding optimal hopping mass [38] and hind limb morphologies resembling quadrupedal marsupials [111], it has been suggested that *Protemnodon* employed a quadrupedal gait, rather than a hopping one [57,111,112]. Humeral morphology in *Protemnodon* also supports this argument indicating a great extent of quadrupedal locomotion when compared to extant kangaroos [58]. Furthermore, Kerr et al. [36] suggested that *Protemnodon* locomotion varied across species residing in different paleoenvironments with a quadrupedal gait in *Protemnodon tumbuna* from the rainforests of New Guinea, low-geared movement in *Protemnodon mankurra* residing in well-wooded regions of southern Australia, and medium-to-high geared with traits convergent to extant *Macropus* in *Protemnodon viator*. Given the nature of sampling, isolated teeth examined here could not be easily identified to species level, so are grouped as local *Protemnodon* from Mt Etna. Locomotion for species of *Protemnodon* from Mt Etna, is poorly understood, however given quadrupedal movements are more economical in a closed forest [113,114], and the tropical rainforest environment reflects rainforest and heavily wooded habitats occupied by low-geared species of *Protemnodon* [36], there is potential limited foraging ranges could be attributed to a lower geared

quadrupedal gait. Associated postcranial fossils from Mt Etna will be needed to further test this hypothesis.

Overall, while most Mt Etna *Protemnodon* occupied small ranges constrained to the Mount Alma Formation, lower $^{87}$Sr/$^{86}$Sr ratios in WIGL8550 suggest that this individual could have occupied a unique foraging range. The mean Sr isotope composition of WIGL8550 (0.706855 ± 0.000020 (2SE)) is relatively similar to bio-available $^{87}$Sr/$^{86}$Sr measured in Alton Downs Basalt (0.706728 ± 0.000022, 2SE, n = 1) occurring ~6 km east and ~7 km southwest, Permian to Triassic Gabbro (0.706839-0.000008, 2SE, n = 1) with small units found <1.5 km south to southwest, and alluvial deposits associated with the Fitzroy River (0.706292 ± 0.000883, 2SE, n = 4) (Figs 1 and 5). As macropodid enamel mineralisation occurs in conjunction with weaning [27,28], $^{87}$Sr/$^{86}$Sr values observed in WIGL8550 could suggest the individual resided over one of these geological substrates during early life. Given the tooth was found in Mt Etna Caves, this small to moderate scale movement (of at least 1.5 km) was undertaken at some point in time, either by the mature *Protemnodon* post-enamel mineralisation, or post-mortem, following transport by a predator.

Strontium isotope ratios that differ from the fossil site's local geology have previously been attributed to predator transport [115]. Fossil deposits at Mt Etna Caves contain several carnivorous taxa including *Thylacinus, Sarcophilus*, and *Thylacoleo* [33,35,40]. Whilst cranial morphology of *Thylacinus*, suggested an inadequacy for handing large-prey items [116,117], *Sarcophilus* are carrion feeders [118,119] suggesting scavenging of deceased *Protemnodon* could provide a viable transport medium for foreign isotope ratios. Additionally, at 100 – 130 kg, *Thylacoleo carnifex* were more than capable of predating on large-bodied macropodids [120]. Given the lack of extant analogous counterparts, little is known about hunting strategies, however, it is plausible that ranges may have extended from Mt Etna Caves to surrounding substrates. Whilst there is no direct evidence of predation on the Mt Etna *Protemnodon*, the presence of *Thylacoleo* and *Sarcophilus* suggests it is conceivable post-mortem transport by a predatory agent could have taken place.

Measurements from WIGL8550 suggest the individual was comparable in size to *P. viator* [36] marginally larger than other individual *Protemnodon* from Mt Etna Caves (Table B in S1 Table). While this may suggest this broader home range to adjacent geological substrates is driven by body mass, smaller foraging ranges, constrained to the Mount Alma Formation are observed in two additional individuals comparable to *P. viator* [36]. Therefore, a larger sample set with both small and large individuals of *Protemnodon* is required to determine whether broader scale dispersal is a product of increased body mass, or simply a unique behaviour observed in some highly vagile individuals. Based on the current dataset, we suggest that most *Protemnodon* from Mt Etna were non-vagile, though, broader scale dispersal to adjacent geological substrates was possible for some individuals.

When comparing these results to previous megafaunal studies, limited dietary ranges have also been suggested for extinct, assemblages of *Diprotodon, Procoptodon* and *Protemnodon* in Bingara and Wellington Caves, New South Wales, Australia [7]. However, in the Darling Downs, Australia, Price et al. [2] suggested that *Diprotodon optatum* were mobile, undertaking annual two-way latitudinal migrations of as much as 200 km. When examining these differences, environments in which megafauna resided in are likely to play a key role in dictating the extent of foraging ranges. As range extent of extant macropodids is strongly associated with climate [15], Koutamanis et al. [7] proposed limited foraging ranges in megafaunal macropodids could be attributed to a similar phenomenon, whereby, macropodids residing in a "rich ecosystem" with relative climatic stability, would be characterised by limited foraging ranges. As discussed above, climatic stability associated with Mt Etna's rainforest environment [33] is likely to have provided necessary primary productivity where dietary needs could be

met with relatively small foraging ranges. Contrasting this, enhanced seasonality at Darling Downs may have been a driving factor behind large-scale, migratory behaviour in *Diprotodon* [2]. These results suggest that the foraging range of Australia's megafauna were complex, where, unlike eutherians, foraging ranges may be more strongly associated with climatic and environmental conditions rather than body size.

## Conclusions

New TT-OSL ages and open system U-Th ages support previous site chronologies established for the Mt Etna caves, and further constrain fossil accumulation in stratigraphic units QML1311H, QML1311C/D, to ~ 280 – 330 ka. Whilst linear regression models predict species of *Protemnodon* should have occupied relatively large ranges, strontium isotope measurements of teeth indicate that *Protemnodon* from Mt Etna Caves had a limited foraging range with movements restricted to the local Mount Alma Formation, with the exception of one specimen originating at least 1 km away. Limited foraging ranges and therefore home range in Mt Etna *Protemnodon* contrast modelled ranges based on body mass. These differences are suggested to have been driven by resource availability, dietary preferences, and/or locomotory biomechanics, that limited the terrestrial dispersal capacity. Limited foraging ranges support conclusions drawn from extant macropodids suggesting foraging ranges may be more strongly associated with climate and environment. There is potential these limiting ranges may have also been a contributing factor to extinction of *Protemnodon* at Mt Etna caves. Localised, restricted, populations of *Protemnodon* may have been well-adapted to a closed-forest environment, however, they were unable to sustain this as intensifying aridification and resource degradation, predisposed them to localised extinction due to their limited dispersal capabilities.

## Supporting information

**S1 Table. Supplementary tables (Table A – Table Q).**
(XLSX)

**S1 File. Supporting information (Text).**
(DOCX)

**S1 Fig. Cross sections of megafauna enamel.** Samples QML1311H-WIGL8543 to QML1312-WIGL8554. Strontium transects can be identified by a series of large depressions running along the enamel/dentine.
(TIF)

**S2 Fig. Stratigraphy of Mt Etna Caves.** Exposed fossil deposits at Mt Etna Limestone Mine, western benches. 1–5 QML1311; 1. A/B, 2. C/D, 3. F, 4. H, 5. J. 6–7 QML1384; 6. LU, 7. UU. 8. QML1310 Unit 2, 9. QML1383 A, 10. Open chamber to Speaking Tub Cave System, 11. QML1313, 12. Bench 0 (A/B), 13. QML1385. Reprinted with permission from Hocknull et al. [33] © 2007 Elsevier B.V. All rights reserved.
(TIF)

**S3 Fig. Single-grain TT-OSL dose-recovery test results.** Radial plot showing the dose-recovery test (natural + dosed) TT-OSL $D_e$ values obtained for sample MTE17-4 after applying the SAR quality assurance criteria. The grey band is centred on the weighted mean $D_e$ value of the unbleached and dosed grains in the dose recovery test, calculated using the central age model (CAM) [86].
(TIF)

**S4 Fig. Representative single-grain TT-OSL decay and dose-response curves for sample MTE17-1.** In the insets, the open circle denotes the sensitivity-corrected natural signal, and filled circles denote the sensitivity-corrected regenerative dose signals. The $D_0$ value characterises the rate of signal saturation with respect to administered dose and equates to the dose value for which the saturating exponential dose-response curve slope is $1/e$ (or ~ 0.37) of its initial value.
(TIF)

**S5 Fig. $^{87}Sr/^{87}Sr$ measured in individual ablation transects undertaken on *Protemnodon* enamel.**
(TIF)

**S6 Fig. Trace element concentrations measured in *Protemnodon* enamel.** Data has been normalised to Ca, and compared to modern Tasmanian bare-nosed wombats reported in Koutamanis et al. [7].
(TIF)

**S7 Fig. Rare earth element concentrations normalised to Ca in enamel for *Protemnodon*.** Results are compared to modern Tasmanian bare-nosed wombats reported in Koutamanis et al. [7].
(TIF)

**S8 Fig. REE concentrations (normalised to Ca) measured in individual ablation spots WIGL8544, WIGL8546, WIGL8547, WIGL8549 and WIGL8550.**
(TIF)

**S9 Fig. REE concentrations (ppm) measured in individual ablation spots for two transects completed in WIGL8546.** Transect 1 (Left) runs from the crown towards the base of tooth and enamel edge (1-23). Transect 2 (right) runs from the base of the enamel towards the crown (1-15).
(TIF)

**S10 Fig. REE concentrations (ppm) measured in individual ablation spots for two transects completed in WIGL8547.** Transect 1 (Left) along the enamel boundary, adjacent to discolouration (1-11). Transect 2 (right) runs alongside to damaged and discoloured enamel (1-15).
(TIF)

**S11 Fig. REE concentrations (ppm) and $^{87}Sr/^{87}Sr$ measured in three individual ablation transects undertaken on *Protemnodon* enamel and dentine for WIGL8546.**
(TIF)

## Acknowledgments

We acknowledge to the Traditional Owners of the Mt Etna region, the Darumbul people, and pay respect to past, present and emerging leaders. SAH wishes to thank Rochelle Lawrence, Noel Sands and staff from Capricorn Caves, for their field and collecting assistance. We thank Cement Australia, Central Queensland Speleological Society, Queensland Parks and Wildlife and Capricorn Caves for their assistance during field-based salvage and stockpiling of the fossiliferous deposits uncovered at Mt Etna. Work undertaken at Mt Etna was by permission and collaboration with Cement Australia and also collaboration with Queensland Parks and Wildlife Service (Scientific Permit WITK17469216). SAH thanks Caitlin Syme for collection management assistance. We would also like to thank Katarina Mikac for reviewing an earlier version of this manuscript.

## Author contributions

**Conceptualization:** Christopher Laurikainen Gaete, Anthony Dosseto, Scott Hocknull.

**Data curation:** Lee Arnold, Martina Demuro, Richard Lewis.

**Formal analysis:** Christopher Laurikainen Gaete, Anthony Dosseto, Lee Arnold, Martina Demuro, Richard Lewis.

**Funding acquisition:** Lee Arnold, Martina Demuro, Scott Hocknull.

**Investigation:** Christopher Laurikainen Gaete.

**Methodology:** Christopher Laurikainen Gaete, Anthony Dosseto, Lee Arnold, Martina Demuro, Richard Lewis, Scott Hocknull.

**Resources:** Scott Hocknull.

**Supervision:** Anthony Dosseto, Scott Hocknull.

**Writing – original draft:** Christopher Laurikainen Gaete.

**Writing – review & editing:** Christopher Laurikainen Gaete, Anthony Dosseto, Lee Arnold, Scott Hocknull.

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
