## [Decision Letter · Decision Letter 0]

17 Sep 2024

PONE-D-24-27519Megafauna mobility: assessing the foraging range of an extinct macropodid, from central eastern Queensland, Australia.PLOS ONE

Dear Dr. Laurikainen Gaete,

Thank you for submitting your manuscript to PLOS ONE. After careful consideration, we feel that it has merit but does not fully meet PLOS ONE’s publication criteria as it currently stands. Therefore, we invite you to submit a revised version of the manuscript that addresses the points raised during the review process.

We look forward to receiving your revised manuscript.

Kind regards,

Julien Louys

Academic Editor

PLOS ONE

Journal Requirements:

2. In your manuscript, please provide additional information regarding the specimens used in your study. Ensure that you have reported human remain specimen numbers and complete repository information, including museum name and geographic location.

For more information on PLOS ONE's requirements for paleontology and archeology research, see https://journals.plos.org/plosone/s/submission-guidelines#loc-paleontology-and-archaeology-research .

4. Please note that funding information should not appear in the Acknowledgments section or other areas of your manuscript. We will only publish funding information present in the Funding Statement section of the online submission form. Please remove any funding-related text from the manuscript.

6. Please amend either the title on the online submission form [(macropodid, from) via Edit Submission] or the title in the manuscript so that they are identical.

7. We note that Figures 1 and 6 in your submission contain map images which may be copyrighted. All PLOS content is published under the Creative Commons Attribution License (CC BY 4.0), which means that the manuscript, images, and Supporting Information files will be freely available online, and any third party is permitted to access, download, copy, distribute, and use these materials in any way, even commercially, with proper attribution. For these reasons, we cannot publish previously copyrighted maps or satellite images created using proprietary data, such as Google software (Google Maps, Street View, and Earth). For more information, see our copyright guidelines: http://journals.plos.org/plosone/s/licenses-and-copyright.

1) You may seek permission from the original copyright holder of Figures 1 and 6 to publish the content specifically under the CC BY 4.0 license. 

2) If you are unable to obtain permission from the original copyright holder to publish these figures under the CC BY 4.0 license or if the copyright holder’s requirements are incompatible with the CC BY 4.0 license, please either i) remove the figure or ii) supply a replacement figure that complies with the CC BY 4.0 license. Please check copyright information on all replacement figures and update the figure caption with source information. If applicable, please specify in the figure caption text when a figure is similar but not identical to the original image and is therefore for illustrative purposes only.

**Additional Editor Comments:**

Both reviewers found your manuscript of great interest, but similar points were raised by both that will need to be addressed before this can be publsihed. Points raised by both reviewers:

1. Your background Sr values need to be derived and calculated from the local Mt Etna region, not similar rock units. Without these, the Sr values from the fossils can't be meaningfully contextualised. This will require additional analytical work.

2. Your interpretations of the palaeobiology of Protemnodon needs to be improved, both in terms of accuracy in our current understanding of the ecological requirements of the genus and constituent species, and the specific individuals found in Mt Etna. This will likely require additional analytical work, or at the very least a thorough and well-reasoned discussion and/or meta-analysis.

In addition, Reviewer 1 has highlighted some significance discrepencies in how your geochronological data was interepreted. This is an important point that will need to be properly addressed. They also pointed out issues with how your data was presented, and emphasised clarity of written expression.

Reviewers' comments:

Reviewer's Responses to Questions

**Comments to the Author**

1. Is the manuscript technically sound, and do the data support the conclusions?

Reviewer #1: No

Reviewer #2: No

2. Has the statistical analysis been performed appropriately and rigorously? 

Reviewer #1: No

Reviewer #2: Yes

3. Have the authors made all data underlying the findings in their manuscript fully available?

Reviewer #1: No

Reviewer #2: Yes

4. Is the manuscript presented in an intelligible fashion and written in standard English?

Reviewer #1: Yes

Reviewer #2: Yes

5. Review Comments to the Author

Reviewer #1: Introduction

Lines 52-54: Mentions a “significant amount of research” but cites none of it. Be sure to add in relevant citations here for studies that both focus on mechanisms, and those that focus on life history. There are several on both accounts, so be sure to cite them.

Lines 56-57: “a large body size provides opportunities for greater geographical range…” greater than what? Need to be clear.

Lines 57-60: “Whilst body mass is considered a strong indicator in placental mammals [5], external factors including habitat type [6] – specifically relationships between opened and closed habitats – may also play a role in dictating the extent of home range.” I’m having trouble understanding the phrasing here. Suggest a re-write to be clear to emphasise exactly what “relationships” you’re talking about.

Lines 60-64: Again, I’m having trouble understanding the arguments here. The paragraph opens with the specific research question about the extinction of Quaternary megafauna, but later deviates to ecological aspects of South American and African extant communities. It’s unclear what the link to late Quaternary extinctions in Australia is here.

Lines 71-76. I’m sorry, but I’m having trouble making of the phrasing through here and the inference with respect to geology, botany, or feeding activities of herbivores.

Lines 78-79: “Strontium isotope ratios in terrestrial consumers are predominately derived from diet.” No, ratios are something that are measured and not derived from diet. I know what you mean, but you need to think about phrasing. Keep in mind too that the accumulation of strontium isotopes in biological tissues is not from ‘diet’, but ‘feeding activities’ (‘diet’ is something we observe but is not the activity itself).

Lines 88-90: Need to clearly state what the “trends” in mineralisation are- i.e., as related to ontogeny.

Lines 100-101: The position of the genus is not poorly understood as implied in this sentence. It’s a sister taxon to Macropus and member of the same family; this is an inference that has been unchallenged for the past 50+ years if not more and is one of the principal findings of the study that is cited here (reference 35- Cascini et al.).

Lines 91-106: This paragraph needs attention. It covers all sorts of themes including introduction of a study site, geography, climate change, extinction, a specific taxon, its taxonomic position, then merging into palaeobiology touching on dentition and post-cranial functional morphology. It’s very hard to follow. Best to think about paragraph structure and their themes.

Line 107: “Given megafaunal macropodids dominate Australia’s fauna…” what does this mean? Macropodids are the largest-bodied endemic herbivores today, but that is not true across geologic time. The largest-bodied mega-herbivores were the diprotodons and ancestors for most of the Quaternary, and all of the 25-odd million years before that.

Materials and Methods

Lines 127-147. You need to cite the primary literature for this source of geological information rather than an unpublished thesis. Most of the synopsis presented in this section does not appear to have come from Deer’s 2011 thesis.

Lines 155-156. The first supplementary table referred to in the manuscript is “Supplementary Table S16”. You need to have data cited in order through the text from 1 to N, not randomly starting at 16. In any case, there is no Table 16 in the Supplementary files- it only goes to S14 so I’m having trouble understanding what you’ve done.

Lines 156-157: “Data were obtained from Tucker et al. [5] and Goldingay [48].” What type of data? Need to be specific if you are using data from multiple sources. Does this pertain to mass and home range?

Looking at Supplementary Table 14, there are numerous and out-dated taxonomic errors in there. If this is the primary source of the manuscript and meant to be “S16”, then it needs to be substantially revised.

Lines 157-159. Need to state what the protocols were, and more specifically, what this means for the submitted study. What is the “weighted average” calculated from?

Lines 159-160. “Phylogeny included in the model...” What model? I’m having trouble understanding the rationale here. A database of extant macropodids was collated (Line 155-157) but the rest of the paragraph does not explain why or how those data were used except for something about an R analysis. You need to be explicit in how you can use this data to make foraging range estimates in a meaningful way.

Lines 165-169. The paper here cites reference 34 (Kerr et al.) as a source of information, but the taxonomy presented in the manuscript does not follow Kerr et al. at all, at least for some of the species mentioned. The authors need to be clear in their methodology.

Lines 173-176. Mentions that some samples were chosen for analysis but doesn’t say what the analysis is. Need to also specifically say what the teeth numbers were, i.e., m1, m2, etc.

------

At this stage, I can’t go much further with this review. I’ve been bogged down in the Introduction and Methods for some time now, and there are numerous things that already need to be addressed for this study to be publishable. I really encourage the authors to spend more time revising the manuscript. I think that there is something really good here and is quite novel, but the significance of the study is lost in the framework as presented. I’m not going to recommend publication at this stage but urge the authors to not give up on what could be an excellent scientific contribution.

For what it’s worth, I’ve briefly read over the rest of the manuscript. Some basic things to note include:

1) Cite and integrate supplementary data properly, and present it logically and in order following the main text.

2) I appreciate that Sr data of local geological formations may not be available, but I strongly do not recommend using Sr values of apparently similar rock types from other continents as substitutes in the construction of an isoscape- this is methodologically unsound for a variety of reasons and ultimately may affect the overall findings (Lines 293-295 and Supplementary Table 5- and noting too, most of the references in this supp. table are to a bibliographic database of the literature, not the actual studies themselves that published the Sr data);

3) You need to better consider the implications of the new ages and not overreach. This has major implications for the study’s findings (e.g., Lines 415-427) that play into aspects of understanding the timing of climate change / habitat loss / extinctions in the region. The new interpretation that the mid-Bruhnes climate change event caused rapid impacts on Mt Etna ecosystems is not supported by the data presented.

a. For example, the oldest previous minimum age for site QML1311H is 454+-48 ka (U-series dated calcite infill of bone), but the burial age from the new luminescence dating is around 100 millennia younger. There must be something wrong somewhere here- do not ignore this discrepancy.

b. For QML1311C/D, the previous maximum age (326+-22 ka from a U-series dated basal flowstone) is consistent with the new burial luminescence age (291+-28.3 ka). The new U-series ages of teeth are much younger (as young as 222+4/-3 ka) but are minimum ages; they are not inconsistent with the burial age, but you then can’t make the argument that the deposit dates to as young as 210 ka (Lines 404-405). There is no evidence for that at all; this is a misinterpretation of the data and premise of the dating approach being that they are not a closed system.

c. For QML1384LU, the youngest minimum age comes from the previously U-series dated calcite infill in bone (332 +-14 ka; although there is a typo in the manuscript where it reports this age as “323”- Line 412). The new U-series dates on teeth are minimum ages (as young as 213+-13/9 ka) and are not inconsistent with the minimum age of the calcite infill, but you can’t then argue that the deposit is as young as 210 ka (Line 416)- this cannot possibly be true. The minimum age of the fossil cannot be younger than the oldest minimum age of the deposit- in this case 332 +-14 ka dated from calcite infill in bone. To argue this is another misinterpretation of the data and approach.

4) The underlying assumption is that species within the study taxon, Protemnodon, are directly analogous, biologically and ecologically speaking, with extant macropodines (e.g., Methods, Results, Discussion). However, that interpretation deviates significantly with current palaeobiological understandings of species within the genus that show that they are unlike anything extant today (Janis et al. 2020 J Mamm. Evol.; Jones et al. 2022 J. Mamm. Evol.; Kerr et al. 2024 Megataxa; papers of which are cited in the manuscript but that general finding is ignored). I strongly encourage the authors to better consider the palaeo-biological/ecological significance of the study taxon in order to best interpret it.

Reviewer #2: This study Is the result of original research and as far as I can discern results have not been reported elsewhere. Methods are well described and rigorously performed to a high standard. The article is intelligible and written in standard English. Data availability guidelines are also met. However, conclusions are not fully supported by the data as follows.

I have two concerns. First, the authors indicate that Sr values associated with Mt Etna Protemnodon suggest short scale movements (Lines 448-449) but “…to reliably estimate the extent of these, a more extensive surveying of bio-available strontium and an isoscape quantifying spatial variability of 87Sr/86Sr will be required” (Lines 452-454). A background Sr dataset is always required for such a study to meet standards of research integrity. Whereas Mt Etna limestone 87Sr/86Sr data were available, using “common…values for corresponding rock types” (Line 294-295) does not meet this standard. A Sr sampling regime that encompasses the estimated home range of Protemnodon based on the taxon’s body mass is needed to support the argument that Protemnodon’s foraging range was smaller than expected, thus not scaled to its body mass but “…likely due to a unique combination of behaviour, diet and/or locomotion” (Lines 40-41, 463-466).

Secondly, the authors argue that Protemnodon was foraging in a “biodiverse rainforest ecosystem” characteristic of Mt Etna prior to 210 ka. Thus, Protemnodon was likely browsing not grazing and a smaller than expected foraging range was not surprising given that “…Mt Etna’s tropical rainforest environment provided a productive ecosystem where dietary needs could be met with relatively small foraging ranges” (Lines 476-477). Yet, no data are provided in support of Protemnodon’s hypothesized dietary intake. Such evidence would be useful adding credence to the author’s explanation for a smaller than expected foraging range relative to body mass. Tooth enamel sampled for Sr yielded very little evidence of diagenetic alteration, (Supplemental Materials.S3). Thus, Protemnodon enamel can also be sampled for hydroxyapatite stable carbon isotope values. Methods are well established and d13C enamel data are commonly considered reliable even in mya settings. If Protemnodon was foraging in a closed canopy rainforest enamel d13C values will be diagnostic of that setting, more negative than expected of a C3 open foraging environment given the recycling of atmospheric CO2. Results will either support or call into questions the authors’ argument that Protemnodon’s foraging range was determined by climate rather than scaled to body mass. A wider sampling of the Sr isoscape is needed to confirm that argument.

6. PLOS authors have the option to publish the peer review history of their article (what does this mean? ). If published, this will include your full peer review and any attached files.

**Do you want your identity to be public for this peer review?** For information about this choice, including consent withdrawal, please see our Privacy Policy .

Reviewer #1: No

Reviewer #2: No

---

## [Author Response · Author response to Decision Letter 0]

16 Oct 2024

All reviewer comments are provided in the attached 'Response to Reviewers' word document.

---

## [Decision Letter · Decision Letter 1]

14 Jan 2025

PONE-D-24-27519R1Megafauna mobility: assessing the foraging range of an extinct macropodid from central eastern Queensland, Australia.PLOS ONE

Dear Dr. Laurikainen Gaete,

Thank you for submitting your manuscript to PLOS ONE. After careful consideration, we feel that it has merit but does not fully meet PLOS ONE’s publication criteria as it currently stands. Therefore, we invite you to submit a revised version of the manuscript that addresses the points raised during the review process.

While both reviewers felt your manuscript was publishable with somewhat minor revisions, these changes will be essential before the manuscript can be accepted. In addition to their comments, there are further moderate revisions that need to be undertaken, as detailed below. 

Reviewer 1 was no longer able to review your MS. Reviewer 2 noted the manuscript overall is solid, however there are some notable discrepencies in your presentation and interpretations that they have highlighted. Firstly, they indicate that there are insufficient details related to the plant sampling and Sr results. I agree with this comment - none of the methodology mentions how the plant Sr values were arrived at, and your discussion pays scant attention to these results. It would also enormously aid the reader if you mapped both the locality and Sr values obtained from the plants on one of your maps. Secondly, their observation that most of your Sr values for Protemnodon have values outside the range of the Mt Etna limestone directly contradicts the substantive claims you make. While some individuals may have been restricted to the Mt Etna limestone, several others clearly were not. This needs to be much better discussed - it doesn't negate all your results but clearly it's more complex than you make out. Reviewer 3 drew attention to your section 197-207. This section is repetitive and confusing - please revise accordingly.

In addition, please address the following:

1. Lines 36-38. I don't think any ecologist would argue that foraging range in eutherians is only dictated by body size, and your results and discussion invalidate this statement in any case (e.g. line 324). Please remove, or reword accordingly.

2. Lines 89-90. This sentence is incomplete.

3. Lines 171-172. You draw attention to the size of one specimen, and clearly the size of the teeth are an important indicator of body size. What about the other teeth? You need to add dimensions of the teeth, or at the very least if this is no longer possible, at least indicate if they likely represent small, medium, or large Protemnodons.

4. Line 191. You've not established that these represent eight individuals, only that they represent eight individual teeth. Please provide molar position for each molar, so you and the reader can determine if there is a possibility that different teeth from the same individual may be present in a unit in Table B, and provide MNIs per unit in the body of the text.

5. Line 204-205. This sentence is incomplete. But also, they don't represent a natural population, which has an ecological definition implying individuals within the population can reproduce. At best, you have individuals of a palaeocommunity.

6. Line 480 - not may, does account for some variation, based on the statistically significant results you reported!

7. Line 483 - first mention of Petrogale so spell out in full. More importantly, how were outliers established? 

8. Line 485 - as above.

9. Lines 517-519. This statement is contradicted by your results (see Reviewer 2 comments). Also, to what does the Funck reference relate to?

10. Lines 524-525. This sentence isn't grammatically correct. More importantly, it's not a fair comparison - one genus of marsupial vs all placentals. I'm sure placentals have their fair share of outliers, as indeed you point out further down.

11. Lines 530-532 - but isn't it analgous to at least P. penicillata? And then in lines 547-550 you rely on them being analgous to form your hypothesis, despite saying they aren't. You need to be internally consistent with your arguments.

12. Line 593 - these animals aren't sedentary, which implies inactive, they just have a small range (although see point 9).

13. Line 618 - what do lemurs have to do with Australian megafauna? No clear link established

Figure 1 - show where in Australia this is. Merge with Figure 6. Add plant localities

Figure 2 - add species labels

We look forward to receiving your revised manuscript.

Kind regards,

Julien Louys

Academic Editor

PLOS ONE

Journal Requirements:

Additional Editor Comments:

While both reviewers felt your manuscript was publishable with somewhat minor revisions, these changes will be essential before the manuscript can be accepted. In addition to their comments, there are further moderate revisions that need to be undertaken, as detailed below.

Reviewer 1 was no longer able to review your MS. Reviewer 2 noted the manuscript overall is solid, however there are some notable discrepencies in your presentation and interpretations that they have highlighted. Firstly, they indicate that there are insufficient details related to the plant sampling and Sr results. I agree with this comment - none of the methodology mentions how the plant Sr values were arrived at, and your discussion pays scant attention to these results. It would also enormously aid the reader if you mapped both the locality and Sr values obtained from the plants on one of your maps. Secondly, their observation that most of your Sr values for Protemnodon have values outside the range of the Mt Etna limestone directly contradicts the substantive claims you make. While some individuals may have been restricted to the Mt Etna limestone, several others clearly were not. This needs to be much better discussed - it doesn't negate all your results but clearly it's more complex than you make out. Reviewer 3 drew attention to your section 197-207. This section is repetitive and confusing - please revise accordingly.

In addition, please address the following:

1. Lines 36-38. I don't think any ecologist would argue that foraging range in eutherians is only dictated by body size, and your results and discussion invalidate this statement in any case (e.g. line 324). Please remove, or reword accordingly.

2. Lines 89-90. This sentence is incomplete.

3. Lines 171-172. You draw attention to the size of one specimen, and clearly the size of the teeth are an important indicator of body size. What about the other teeth? You need to add dimensions of the teeth, or at the very least if this is no longer possible, at least indicate if they likely represent small, medium, or large Protemnodons.

4. Line 191. You've not established that these represent eight individuals, only that they represent eight individual teeth. Please provide molar position for each molar, so you and the reader can determine if there is a possibility that different teeth from the same individual may be present in a unit in Table B, and provide MNIs per unit in the body of the text.

5. Line 204-205. This sentence is incomplete. But also, they don't represent a natural population, which has an ecological definition implying individuals within the population can reproduce. At best, you have individuals of a palaeocommunity.

6. Line 480 - not may, does account for some variation, based on the statistically significant results you reported!

7. Line 483 - first mention of Petrogale so spell out in full. More importantly, how were outliers established?

8. Line 485 - as above.

9. Lines 517-519. This statement is contradicted by your results (see Reviewer 2 comments). Also, to what does the Funck reference relate to?

10. Lines 524-525. This sentence isn't grammatically correct. More importantly, it's not a fair comparison - one genus of marsupial vs all placentals. I'm sure placentals have their fair share of outliers, as indeed you point out further down.

11. Lines 530-532 - but isn't it analgous to at least P. penicillata? And then in lines 547-550 you rely on them being analgous to form your hypothesis, despite saying they aren't. You need to be internally consistent with your arguments.

12. Line 593 - these animals aren't sedentary, which implies inactive, they just have a small range (although see point 9).

13. Line 618 - what do lemurs have to do with Australian megafauna? No clear link established

Figure 1 - show where in Australia this is. Merge with Figure 6. Add plant localities

Figure 2 - add species labels

Reviewers' comments:

Reviewer's Responses to Questions

**Comments to the Author**

1. If the authors have adequately addressed your comments raised in a previous round of review and you feel that this manuscript is now acceptable for publication, you may indicate that here to bypass the “Comments to the Author” section, enter your conflict of interest statement in the “Confidential to Editor” section, and submit your "Accept" recommendation.

Reviewer #2: (No Response)

Reviewer #3: (No Response)

2. Is the manuscript technically sound, and do the data support the conclusions?

Reviewer #2: Yes

Reviewer #3: Yes

3. Has the statistical analysis been performed appropriately and rigorously? 

Reviewer #2: Yes

Reviewer #3: Yes

4. Have the authors made all data underlying the findings in their manuscript fully available?

Reviewer #2: Yes

Reviewer #3: Yes

5. Is the manuscript presented in an intelligible fashion and written in standard English?

Reviewer #2: Yes

Reviewer #3: Yes

6. Review Comments to the Author

Reviewer #2: Once again, this study Is the result of original research and as far as I can discern results have not been reported elsewhere. Methods are well described and rigorously performed to a high standard. The article is intelligible and written in standard English. Data availability guidelines are also met. However, I am puzzled by seemingly contradictory narratives even though the manuscript has been significantly revised and new temporal data added.

First, the authors note that 24 local plants were sampled for Sr, in an attempt to meet the need for a background Sr dataset suggested in previous reviews. Values are reported in S1_Table, ranging from 0.705817-0.709395, a much wider range of Sr values than those seen in Protemnodon. These data appear to support the authors’ argument but are not discussed in the manuscript. Were these samples collected within the foraging range of Protemnodon based on estimates of body mass or within the narrower foraging range proposed by the authors? The plant study is mentioned in Lines 314-316 only. A more in-depth treatment of these data would be useful.

I am also puzzled by conclusions the authors’ reach based on S5_Fig. This figure shows that a substantial subset of measured Sr values falls outside the Mt Etna caves’ range in five of the eight teeth sampled, given that dotted lines mark the range of Sr values on Mt Etna limestone (Line 410-412). Rather than fully acknowledging this finding the authors argue that most sampling units overlap those of Mt Etna limestone attributing the bulk of the variability to a single dental sample with Sr values entirely outside this range (Line 46-418). A more conservative interpretation of reported Sr data would argue that whereas four of the individuals sampled appear to have foraged within a narrow Mt Etna home range, others foraged more widely. Given a relatively wide range of body mass estimates for this taxon (50-170kg, Lines 325-328), this finding seems reasonable. U-Th ages reported in Table 2 do not suggest that these differences are temporally patterned. Thus, variability in foraging behavior seems a parsimonious interpretation of the data.

Moreover, if Protemnodon was foraging in a closed canopy rainforest enamel d13C values will be diagnostic of that setting, more negative than expected of a C3 open foraging environment given the recycling of atmospheric CO2. Results will either support or call into question the authors’ argument that Protemnodon’s foraging range was narrow and determined by climate rather than scaled to body mass. This is a straightforward analytical procedure, well established methodologically. It was suggested in a previous review but not conducted or reported here.

Reviewer #3: The study presents novel and original research into the mobility and foraging ranges of mid-Pleistocene Mt Etna Protemnodon populations, as well as further dating constraints on two Mt Etna stratigraphic units. Overall, the paper is written succinctly, with a detailed methodology outlining how the study was conducted. Moreover, the data presented in the paper appears to support the conclusions made by the authors. Measured Sr values within the teeth appear to be similar to the bio-available strontium levels within the Mt Etna region, suggesting that the local Protemnodon had limited foraging ranges. The dates also appear to be consistent with previously published data on the sites.

Lines 197-207: This section is difficult to follow. I think the authors are saying that: 1) the selected teeth could be assigned to the genus Protemnodon, but not to a species; and 2) the dataset represents home ranges and mobility of mid-Pleistocene Mt Etna Protemnodon populations, not the genera as a whole. This section should be written more succinctly.

Lines 502-506 and 536-539: These sections use two different listing styles (“1”, “2” and then “i”, “ii”, “iii”). I recommend choosing one style and keeping it consistent throughout the paper.

7. PLOS authors have the option to publish the peer review history of their article (what does this mean? ). If published, this will include your full peer review and any attached files.

**Do you want your identity to be public for this peer review?** For information about this choice, including consent withdrawal, please see our Privacy Policy .

Reviewer #2: No

Reviewer #3: No

---

## [Author Response · Author response to Decision Letter 1]

30 Jan 2025

See ' Response to Reviewers.doc' attached.

---

## [Editor Report · Decision Letter 2]

6 Feb 2025

Megafauna mobility: assessing the foraging range of an extinct macropodid from central eastern Queensland, Australia.

PONE-D-24-27519R2

Dear Dr. Laurikainen Gaete,

We’re pleased to inform you that your manuscript has been judged scientifically suitable for publication and will be formally accepted for publication once it meets all outstanding technical requirements.

Kind regards,

Julien Louys

Academic Editor

PLOS ONE
---

## [Editor Report · Acceptance letter]

PONE-D-24-27519R2

PLOS ONE

Dear Dr. Laurikainen Gaete,

I'm pleased to inform you that your manuscript has been deemed suitable for publication in PLOS ONE. Congratulations! Your manuscript is now being handed over to our production team.

Kind regards,

on behalf of

Dr. Julien Louys

Academic Editor

PLOS ONE